# Characterization and Analysis of the Complete Mitochondrial Genome of *Platycrater arguta*

**DOI:** 10.3390/cimb47070521

**Published:** 2025-07-05

**Authors:** Xule Zhang, Lei Feng, Xiaohua Ma, Qingdi Hu, Yaping Hu, Jian Zheng

**Affiliations:** Key Laboratory of Plant Innovation and Utilization, Institute of Subtropical Crops of Zhejiang Province, Wenzhou 325005, China; zhangxl@zaas.ac.cn (X.Z.); fengl@zaas.ac.cn (L.F.); maxh@zaas.ac.cn (X.M.); huqd@zaas.ac.cn (Q.H.); huyp@zaas.ac.cn (Y.H.)

**Keywords:** mitogenome, *Platycrater arguta*, comparative genomics, phylogeny

## Abstract

*Platycrater arguta* (Hydrangeaceae), a rare and endangered Tertiary relict shrub endemic to East Asia, holds significant ecological and evolutionary value. However, the *P. arguta* mitochondrial (mt) genome remains unexplored, limiting insights into its cytoplasmic evolution and phylogenetic relationships. In this study, a complete mt genome of *P. arguta* was sequenced, and we assembled the mt genome into two linear contigs for description, due to the complexity of its chromosome structure. The mt genome encodes 37 protein-coding genes, 27 tRNA genes, and three rRNA genes. A total of 687 RNA editing sites were predicted, and the most RNA editing sites were found in the *nad4* gene. Repeat sequences with different sizes were detected in the mt genome, including 160 simple sequence repeats, 26 tandem repeats, and 320 dispersed repeats. Phylogenetic analysis grouped *P. arguta* with *Hydrangea macrophylla* (Hydrangeaceae), which is closely related to Eucommiaceae and Ericaceae. This study provides the first assembled and annotated mt genome of *P. arguta*, which enhances our understanding of the genome of this relict plant in Hydrangeaceae. Taken together, this study offered foundational data for conservation strategies, molecular breeding, and evolutionary studies of this endangered relict species.

## 1. Introduction

*Platycrater arguta*, a rare and endangered montane deciduous shrub endemic to East Asia, is discontinuously distributed in montane vegetation of eastern China (Fujian, Jiangxi, Zhejiang) and southern Japan, and is of the family Hydrangeaceae [1]. As a Tertiary relict species, *P. arguta* is monotypic with two varieties (*var. sinensis* in China and *var. arguta* in Japan), which hold immense phylogeographic and evolutionary significance [2,3]. The species is faced with serious threats of habitat fragmentation, low seed germination percentage (<40% under optimal conditions), pollen limitation, and human disturbance, and, thus, is proclaimed a nationally protected plant in China [4,5]. Apart from these ecological values as a paleo-endemic relic, *P. arguta* is prized for its ornamental value, medicinal value, and possesses complex spider web-like bracts and double floral forms (bisexual and male flowers), along with pharmacological activity owing to bioactive compounds such as iridoids, flavonoids, and lignans [6].

Mitochondria, which are semi-autonomous organelles, are pivotal in plant stress adaptation by regulating energy metabolism, eliminating reactive oxygen species (ROS), and programmed cell death processes that are critical for survival during environmental stress like drought and light stress [7]. Since the first *Arabidopsis thaliana* mitochondrial genome (mt genome) sequencing in 1997, mt genomic studies have enhanced our understanding of cytoplasmic inheritance, structural variation, and evolutionary relationships in horticultural crops and species like *Camellia sinensis* and *Rhododendron simsii* [8,9]. Unlike highly conserved chloroplast genomes (cp genomes), plant mt genomes are very plastic in size (66 kb–11.7 Mb), gene complement, and recombination rate, and they contain beneficial information about phylogenetic divergence and speciation [10]. The latest cp genome sequencing of *P. arguta* (157,810 bp) established close phylogenetic affinities with *Schizophragma hydrangeoides* [11], while its mt genome has not been characterized yet, limiting in-depth investigations of organelle–nuclear interactions and adaptive evolution in this critically threatened plant.

Research on *P. arguta* has been focused on its reproductive ecology, physiological response to abiotic stress factors, and conservation strategies. Current research reveals that this species exhibits intrinsically low regeneration (primarily due to seed dormancy), stringent germination conditions (e.g., gibberellin supplementation, 25 °C optimum), and obligate outcrossing mediated by specialist pollinators (e.g., bumblebees) [1,12,13]. Drought and light intensity experiments further demonstrate sensitivity towards water stress as well as shade adaptation by physiological adjustment through photosynthesis regulation, antioxidant enzyme activity (e.g., SOD, CAT), and biomass allocation [14]. Molecular analyses with cpDNA and nuclear genes have revealed rich genetic diversity within fragmented populations (*hT* = 0.88 for cpDNA) and deep divergence between Chinese and Japanese types (*FST* = 0.915), and this may be due to Quaternary climatic oscillations and geographic separation [15,16]. However, the lack of mt genomic data prevents a full understanding of its cytoplasmic genetic structure and evolutionary history.

Here, we genome-assembled and whole-genome sequenced the complete mt genome of *P. arguta* using Illumina and Nanopore platforms. We analyzed its structural features (e.g., repeat elements, codon usage, sites of RNA edits) and conducted comparative genomics to investigate evolutionary relationships in Hydrangeaceae. This research not only fills an essential gap in relict species organelle genomics but also provides a basis for the development of better conservation strategies, elucidating stress adaptation mechanisms, and the direction of molecular breeding operations for this rare plant.

## 2. Materials and Methods

### 2.1. Plant Materials and DNA Sequencing

Fresh leaf tissues of *P. arguta* were provided by Dr. Zheng and cultivated under natural conditions at the Zhejiang Institute of Subtropical Crops (Ouhai District, Wenzhou, China; latitude 27°99′ N, longitude 120°63′ E). Fresh tissue samples were frozen in liquid nitrogen immediately after collection and then stored in a −80 °C refrigerator to maintain DNA integrity. Total DNA was extracted from fresh samples using the DNeasy Plant Mini Kit (Qiagen, Valencia, CA, USA), and selected DNA samples were subjected to third-generation sequencing using the HiFi mode of the PacBio SequeII platform. Perl scripts were employed for statistical analysis of the third-generation sequencing data.

### 2.2. Genome Assembly and Annotation

The HiFi raw data were assembled using PMAT2 (v2.0.2), with the assembly visualized and manually refined via Bandage (v0.8.1) to generate a draft assembly. The HiFi reads were then mapped to this assembly using minimap2 and polished with NextPolish, yielding the final assembly sequence with corresponding GFA file adjustments.

Protein-coding genes and rRNAs were annotated by aligning sequences against published plant mitochondrial reference genomes using BLAST+ (v2.16.0), followed by manual refinement based on close relatives. tRNAs were identified using tRNAscan-SE [17] (https://lowelab.ucsc.edu/tRNAscan-SE/index.html, accessed on 23 February 2025). ORFs (Open Reading Frames) were annotated with ORF Finder (http://www.ncbi.nlm.nih.gov/gorf/gorf.html, accessed on 23 February 2025) using a minimum length cutoff of 102 bp; sequences overlapping known genes or showing redundancy were excluded. ORFs > 300 bp were functionally annotated by alignment against the nr database. RNA editing sites were predicted by using PmtREP (http://112.86.217.82:9919/#/tool/alltool/detail/336, accessed on 23 February 2025). Following manual curation and verification of the initial results, the final annotation set was obtained.

### 2.3. Relative Synonymous Codon Usage (RSCU) Analysis

We performed relative synonymous codon usage (RSCU) analysis to assess codon preference in the mt genome of *P. arguta.* A custom Perl script was employed to extract unique protein-coding sequences (CDSs) and quantify the frequency of synonymous codon utilization.

### 2.4. Repeat Sequences Analysis

To characterize repetitive elements in the mt genome, we conducted a comprehensive analysis of dispersed repeats, tandem repeats, and simple sequence repeats (SSRs). Dispersed repeats, including forward, reverse, complement, and palindromic repeats, were identified using BLASTn (v2.10.1) with stringent parameters (word_size 7, e-value ≤ 1 × 10^−5^). Tandem repeats were identified using TRF v4.09 [18] with the parameters “2 7 7 80 10 50 500 -f -d -m”. For microsatellite analysis, MISA (v2.1) was employed to locate SSRs with motif thresholds set at 10, 5, 4, 3, 3, and 3 repeats for mono- to hexanucleotide motifs, respectively, while ensuring a minimum inter-SSR distance of 100 bp. The distribution and organization of repeat elements were visualized using the Circos package in TBtools (v2.003).

### 2.5. Identification of RNA Editing Sites

RNA editing sites in protein-coding genes (PCGs) were predicted using the PmtREP online tool (http://cloud.genepioneer.com:9929/#/tool/alltool/detail/336), with a confidence threshold set at 0.2. We conducted comparative analysis across 10 Malpighiales species, quantifying both the distribution and density (sites/kb) of editing sites to investigate evolutionary patterns of RNA editing. The results were visualized through a stacked bar chart and a normalized heatmap, both generated using Origin 2018 [19].

### 2.6. DNA Transfer Between the Chloroplast and Mitochondrion

Comparative genomic analysis was performed using BLAST alignment in TBtools v2.003 to investigate potential sequence transfer between the chloroplast and mt genomes. For *P. arguta*, the mt genome was compared against its chloroplast counterpart with stringent parameters (identity ≥ 70%, e-value ≤ 1 × 10^−5^, alignment length ≥ 30 bp) [20]. The identified homologous sequences, indicative of possible inter-organellar DNA transfer events, were visualized using the Circos package within TBtools.

### 2.7. Ka/Ks and Pi Analysis

We grouped the advanced analyzed species two by two and extracted the homologous gene pairs, then we used the mafft v7.427 (https://mafft.cbrc.jp/alignment/software/) software to compare the homologous gene pairs, and, after alignment, the KaKs_Calculator v2.0 (https://sourceforge.net/projects/kakscalculator2/) software was used to calculate the Ka and Ks values of each gene pair, and the calculation method was selected as MLWL. Finally, the ka/ks values of each gene pair were counted. Global comparison of homologous gene sequences from different species was performed using the mafft software (v7.427, --auto mode), and Pi values for each gene were calculated using dnasp5.

### 2.8. Phylogenetic Analyses

A maximum likelihood evolutionary tree was made using CDS, inter-species sequences were compared with multiple sequences using mafft software (v7.427, --auto mode), the compared sequences were joined head to tail and trimmed with trimAl (v1.4.rev15) (parameter: -gt 0.7), model prediction was performed using the jmodeltest-2.1.10 software after trimming to determine that the model was of GTR type, and then the maximum likelihood evolutionary tree was constructed using the RAxML v8.2.10 (https://cme.h-its.org/exelixis/software.html) software with GTRGAMMA model selected and bootstrap = 1000 [21].

## 3. Results

### 3.1. General Features of the P. arguta Mt Genome

The mt genome of *P. arguta* was sequenced, assembled, and annotated. A total of 10,396,477,834 pieces of raw data and 591,417 bp of clean data were obtained using the PacBio Sequel II platform. The subreads were N50 and the mean reads were 17,594 bp and 17,578 bp, respectively (Appendix A). Upon addressing the repeat regions, a complex assembly graph of the mt genome was established (Figure 1). Apparently, the mt genome of *P. arguta* adopts a unique conformation with three pairs of repeats. Next, we solved these repeats by artificially simulating possible paths and making judgments based on the mapping results of long reads (Appendix A). So, we obtained a total of 15 contigs by merging redundant nodes and a complex multibranched conformation. Despite efforts to reconstruct mtDNA as a closed-loop or a single linear molecule via standard methods, branches persistently prevented its simplification into a unified form. Hence, we proceeded the mtDNA into two linear molecules in the order of contig 9-contig11-contig5-contig15-contig2-contig7-contig11-contig3 and contig1-contig14-contig10-contig4-contig15-contig12-contig13-contig8-contig8-contig14-contig6, respectively (Appendix A, Appendix A). Of course, it needs to be explained that the treatment here is not the only form due to the complexity and dynamics of plant mtDNA configuration, and the treatment here was selected since it was convenient for subsequent analysis. Hence, according to the treatment, the lengths of contig 1 and 2 were 272,565 bp and 249,405 bp with GC contents of 45.66% and 45.58%, respectively (Figure 2, Appendix A). The mt genome assembly exhibits high quality, a gap-free structure, and a coverage depth of 197.5× (Appendix A).

There were 68 genes annotated in the mt genome of *P. arguta*, including 37 PCGs, 27 tRNA genes, three rRNA genes, and 1 pseudo gene. The 33 mt PCGs, divided into 10 classes, included five ATP synthases (*atp*), four cytochrome c biogenesis proteins (*ccm*), one ubiquinol cytochrome c reductase (*cob*), three cytochrome c oxidases (*cox*), one maturase (*mat*), one transport membrane protein (*mtt*), nine NADH dehydrogenases (*nad*), three ribosomal large subunits (*rpl*), eight ribosomal small subunits (*rps*), and two succinate dehydrogenases (*sdh*) (Table 1). Among them, eight genes, namely, *ccmFc*, *cox2*, and *rps3,* included one intron, genes of *nad1* and *nad2* contained two introns, genes of *nad4* and *nad5* contained three introns, and one gene of *nad7* had four introns (Table 1). Among 27 tRNA genes, four tRNA genes were identified in two copies (*trnF-GAA*, *trnK-TTT*, *trnP-TGG,* and *trnT-TGT*), and *trnM-CAT* was identified in four copies (Figure 2, Table 1).

### 3.2. Codon Preference of the Mitogenome

The codon usage bias of the 37 PCGs in the mt genomes of *P. arguta* was analyzed. Among them, the majority initiated translation with ATG start codons. Notably, only the *cox1* and *nad4L* genes exhibited ACG as their start codon, a feature potentially attributable to C-to-U RNA editing at the second nucleotide position. Five types of stop codons, including TAA, TGA, TAG, CAA, and CGA, were found in the PCGs, and the C to U for RNA editing phenomenon was discovered in *atp6*, *atp9*, *ccmFc*, *and sdh4* (Appendix A). We also calculated the relative synonymous codon usage (RSCU) of 37 PCGs in the *P. arguta* mt genome (Figure 3). The 37 PCGs collectively spanned 31,308 bp, encoding 10,436 codons including stop codons. Among all encoded amino acids, leucine (Leu, 1070 codons; 10.25%) exhibited the highest frequency, followed by serine (Ser, 988 codons; 9.47%). In contrast, termination codons (Ter, 37 codons) represented the least abundant category, comprising only 0.35% of the total codon usage, and methionine (Met) and tryptophan (Trp) were the least utilized amino acids among the 37 PCGs. Apart from the initiation codon (AUG) and the tryptophan codon (UGG), both of which had an RSCU value of one, the mitochondrial PCGs exhibited a strong A/U bias at the third codon position in the *P. arguta* mt genome. Among 29 codons with RSCU values > 1, the majority (27 codons, 93%) terminated with A or U, while only one codon each (3.45%) ended with G or C, demonstrating a pronounced preference for A/U-ending codons (Appendix A).

### 3.3. Repeat Sequence Analysis of the Mitogenome

The repeat sequences consist of SSRs, tandem repeats, and dispersed repeat sequences. In this study, a total of 160 SSRs were found in the *P. arguta* mitogenome, including 49 mono-, 38 di-, 11 tri-, 52 tetra-, 8 penta-, and 2 hexa-nucleotide repeats (Table 2, Figure 4). Approximately 50% of the SSRs are composed of monomers and dimers, with A/T and AT/TA representing the most predominant monomer and dimer types, respectively. Interestingly, the tetramers accounted for 32.5% of the SSRs, which was higher than the proportion of other SSRs, while tri-, penta-, and hexa-nucleotide repeats were fewer in number. Additionally, 26 tandem repeats with a percent match of more than ≥75% and a length ranging from 5 bp to 39 bp were also detected in the mitogenome of *P. arguta* (Figure 4, Appendix A). There was a total of 320 dispersed repeats (≥29 bp), of which 168 palindromic (52.5%) and 152 forward repeats (47.5%) were observed, and no reverse and complementary repeats were found (Figure 4, Appendix A). The total length of the dispersed repeats was 79,506 bp, which occupied 15.23% of the whole mt genome. Most repeats were 29–100 bp (266 repeats, 83.125%), while only two dispersed repeats were over 1 kb, being 2149 bp and 12,915 bp, respectively (Appendix A).

### 3.4. Analysis of Homologous Fragments of Mitochondria and Chloroplasts

The *P. arguta* mitogenome sequence was approximately 3.3 times longer than its cp genome (157,812 bp). The homologous fragments, ranging from 30 bp to 13,000 bp, were identified by sequence similarity analysis. As shown in Figure 5 and Appendix A, a total of 31 fragments with a length of 31,177 bp migrated from the cp genome to the mt genome of *P. arguta*, accounting for 5.97% of the mt genome. Twelve annotated genes wholly located on these fragments, namely, *trnT-TGT*, *trnF-GAA*, *trnM-CAT*, *trnV-GAC*, *trnI-GAT*, *trnA-TGC*, *trnR-ACG*, *trnN-GTT*, *trnW-CCA*, *trnP-TGG*, *trnD-GTC*, and *trnH-GTG*, were likely to have originated from the cp genome. Only partial sequences of 10 PCGs, including *rps4*, *rpl16*, *rrn16*, *rrn23 accD*, *psaA*, *ndhF*, *psbC*, *ycf2*, and *ycf3*, migrated from the cp genome to the mitogenome (Appendix A). This demonstrates that, during inter-organellar DNA fragment transfer in *P. arguta*, tRNA genes exhibit higher evolutionary conservation than PCGs.

### 3.5. The Prediction of RNA Editing Sites in PCGs

The mt genome of *P. arguta* was found to harbor 687 predicted RNA editing sites across all 37 PCGs, consistent with the widespread occurrence of RNA editing events in plant mt genomes (Table 3). Among these PCGs, *nad3* had one RNA-editing site, whereas the highest was in *nad4,* with 54 RNA-editing sites, of which 35.22% (242 sites) occurred at the first position of the triplet codes and 64.77% (445 sites) were located at the second base of the triplet codes (Appendix A). In addition, the first and second bases of the triplet codes were edited, leading to an amino acid change from proline (CCC) to phenylalanine (TTC). There were 33.33% (229 positions) of amino acids whose hydrophobicity remained unchanged after the RNA editing. Additionally, 46.87% (322 positions) of the amino acids varied from hydrophilic to hydrophobic, while 8.15% (56 positions) ranged from hydrophobic to hydrophilic. Furthermore, 0.87% (6 positions) of the amino acids varied from hydrophilic to a stop codon. The findings in our study showed that most amino acids were changed from serine to leucine (21.25%, 146 sites), proline to leucine (19.51%, 134 sites), and serine to phenylalanine (16.60%, 114 sites) (Table 3).

### 3.6. Ka/Ks and Pi Analysis Reveals Selection Pressures

The non-synonymous-to-synonymous substitution ratio (Ka/Ks) is important in ge- netics for assessing the magnitude and direction of natural selection acting on homologous genes among divergent species. We conducted comparative Ka/Ks analysis of 37 mitochondrial PCGs using *P. arguta* as the reference species to assess evolutionary selection pressures (Figure 6). Most PCGs exhibited Ka/Ks ratios < 1, indicating the stability of the protein function of these genes during evolution. However, the Ka/Ks values of the *atp4* (1.05), *ccmB* (1.39), and *nad4* (1.15) were higher than 1, which means that these PCGs were subject to positive selection. The nucleotide diversity (Pi) values of 37 PCGs were accounted for and varied from 0.01782 to 0.10852, with an average of 0.04719. The Pi value of gene *atp9* was largest among these regions, being 0.10852 and 0.01782 in gene *rps14*, 0.02399 in gene *nad7*, 0.02442 in gene *nad4L,* and 0.02487 in gene *nad9*. The lower Pi values revealed that the mt genome sequences of *P. arguta* were highly conserved (Appendix A, Appendix A).

### 3.7. Phylogenetic Analysis

To determine the evolutionary status for the mitogenome of *P. arguta*, the phylogenetic analyses were carried out on *P. arguta* together with 34 other species, including 33 plants from Asteranae and 1 plant from Rosanae (designated as the outgroup) (Figure 7). Due to the limited availability of publicly accessible mt genome data from species within the same genus, family, or even order as *P. arguta*, our phylogenetic analysis was constrained to incorporate taxa within the same superorder (Asteranae) for comparative evolutionary inference. Then, a phylogenetic tree was constructed using an aligned data matrix of 37 conserved PCGs from all tested species. The phylogenetic tree shows that the 35 species were divided into eight groups. *P. arguta* was classified in Group V, which also includes *Hydrangea macrophylla* (a plant from the same family as *P. arguta*). In the phylogenetic tree, the closest relatives to Group V are Group IV (containing *Eucommia ulmoides*) and Group VI (containing *Rhododendron simsii*). This suggests that, in mitochondrial evolution, the Hydrangeaceae family (to which *Platycrater* belongs) may share a closer phylogenetic relationship with Eucommiaceae and Ericaceae.

## 4. Discussion

### 4.1. Characterization of the P. arguta Mitogenome

Mitochondria serve as the central energy-producing organelles in eukaryotes, and plant mt genomes are notably more intricate than their animal counterparts due to dynamic size fluctuations and extensive repetitive sequences [22,23]. In this study, we characterized the mt genome of *P. arguta*, which spans 272,565 bp and is shorter than that of *H. macrophylla* (599,499 bp). The mt genome adopts a complex structure, a common feature among plant mt genomes. This genome size is relatively compact, suggesting lineage-specific evolutionary constraints on genome expansion. Non-coding regions, a hallmark of plant mt genomes, likely constitute a significant portion of the *P. arguta* genome, as evidenced by the presence of 8–10 introns across genes, such as *nad7* (four introns) and *nad4* (three introns), alongside extensive intergenic spacers. This structural complexity mirrors patterns observed in *A. thaliana* and other angiosperms [24]. PCGs accounted for approximately 28.3% of the genome, with most initiating at the canonical ATG codon and terminating at TAA or TGA. Notably, *cox1* utilized ACG as a putative start codon, a phenomenon previously documented in *Salix suchowensis* and *Phaseolus vulgaris* [25,26], likely attributable to RNA editing, a widespread post-transcriptional modification in plant mitochondria. The tRNA repertoire included duplicated genes (*trnT-TGT*, *trnM-CAT*, *and trnF-GAA*), which may enhance translational robustness or mitigate mutational load, as reported in *A. thaliana* [27]. Intriguingly, intron distribution varied markedly among respiratory complex genes; *nad1* and *nad2* harbored two introns, while *nad7* contained four, reflecting divergent evolutionary trajectories in intron retention or loss. These features, coupled with the genome’s moderate size, underscore a balance between structural conservation and plasticity, potentially driven by repetitive element dynamics [28,29]. Further comparative studies within Hydrangeaceae could elucidate lineage-specific adaptations in mt genome architecture and functional innovation.

### 4.2. Mitogenome Revealed Distinctive Molecular Evolutionary Features

Codon usage bias, the non-random selection of synonymous codons across species, reflects evolutionary pressures such as natural selection, mutation bias, and genetic drift [30,31]. In this study, we analyzed the relative synonymous codon usage (RSCU) in the *P. arguta* mt genome, revealing a pronounced A/U bias at the third codon position. Among 29 codons with RSCU values > 1, 93% terminated with A or U. This preference likely stems from mutation pressure favoring AT-rich regions and selection for translational efficiency in plant mitochondria [32]. Leucine (Leu, 10.25%) and serine (Ser, 9.47%) emerged as the most abundant amino acids, consistent with findings in *Suaeda glauca* and *Brassica napus* [33], suggesting conserved metabolic demands for these residues in respiratory complexes. Notably, termination codons (Ter) were rare (0.35%), with UAA predominating (19 occurrences). Intriguingly, *cox1* and *nad4L* utilized ACG as start codons, a feature potentially corrected to ATG via C-to-U RNA editing, as reported in *Phaseolus vulgaris* and *Salix suchowensis* [34]. Such editing events may mitigate genomic constraints while maintaining functional protein synthesis. These findings underscore the interplay between mutational bias and selective forces in shaping codon usage, reinforcing the evolutionary stability of A/U-rich regions in plant mt genomes. Further comparative studies across Hydrangeaceae could elucidate lineage-specific adaptations in translational machinery.

The frequency and distribution of RNA editing sites in plant mt genomes vary significantly across species, with angiosperms often exhibiting extensive editing events [35]. In this study, we identified 687 RNA editing sites across all 37 PCGs in the *P. arguta* mt genome, a number exceeding those reported in *Diospyros oleifera* (515), *Bupleurum chinense DC* (517), and *Pereskia aculeata* (362), but comparable to *Macadamia integrifolia* (688) [8]. This high editing count underscores the dynamic nature of post-transcriptional modifications in *P. arguta.* RNA editing in *P. arguta* predominantly targeted the second position in the codon (64.77%, 445 sites), followed by the first position (35.22%, 242 sites), with no editing detected at the third position—a pattern consistent with *Brassica rapa* and *Salix suchowensis* [36]. Notably, 46.87% of edits altered amino acids from hydrophilic to hydrophobic, potentially enhancing protein stability in hydrophobic membrane environments. Such compositional bias toward C-to-U editing mirrors trends in other plant mt genomes, suggesting conserved molecular mechanisms [37]. Intriguingly, 0.87% of edits introduced premature stop codons, which may regulate gene expression or produce truncated proteins with specialized functions. The prevalence of proline-to-phenylalanine edits further highlights the role of RNA editing in diversifying protein function without altering genomic sequences. The extensive RNA editing in *P. arguta* reflects both evolutionary conservation and lineage-specific adaptations, driven by selective pressures to optimize mitochondrial function.

Repetitive sequences, including tandem and dispersed repeats, are ubiquitous in plant mt genomes and play critical roles in genomic recombination and structural plasticity [38,39]. In this study, we identified 26 tandem repeats (5–39 bp) and 320 dispersed repeats (≥29 bp) in the *P. arguta* mt genome. Dispersed repeats spanned 129,914 bp, accounting for 15.32% of the genome, a proportion significantly higher than that reported in *Brassica oleracea* var. *Italica* (15%), suggesting lineage-specific expansion of repetitive elements [40]. Notably, one exceptionally large repeat (15,064 bp) was identified, which may act as a recombination hotspot, potentially contributing to the genome’s structural dynamism [41]. The majority of dispersed repeats (69.06%) fell within the 29–100 bp range, where short repeats facilitate frequent recombination events. Palindromic repeats often mediate intramolecular recombination, a mechanism critical for maintaining genome stability amid structural rearrangements [42]. The absence of reverse and complementary repeats in *P. arguta* contrasts with findings in *A. thaliana*, possibly reflecting evolutionary constraints on repeat diversification in Hydrangeaceae.

The Ka/Ks ratio serves as a critical indicator of evolutionary selection pressures, with values < 1 reflecting purifying selection and >1 suggesting positive selection [43]. In this study, most mitochondrial PCGs in *P. arguta* exhibited Ka/Ks ratios < 1. However, positive selection (Ka/Ks > 1) was detected in several genes across interspecific comparisons. Nucleotide diversity (Pi) analysis further revealed conserved and variable regions within the *P. arguta* mt genome. The *atp9* gene displayed the highest Pi value (0.10852), suggesting its potential utility as a molecular marker for population genetics, akin to the variable *ccmB* and *rps1* regions in *Phaseolus vulgaris* [8]. In contrast, low Pi values in *rps14* (0.01782), *nad7* (0.02399), and *nad4L* (0.02442) underscored strong functional constraints on these genes. Notably, *rpl10* exhibited moderate diversity (Pi = 0.05168), a feature retained in most plant mt genomes except some Brassicaceae species, where it is replaced by nuclear-derived homologs [44]. These findings emphasize the balance between sequence conservation and adaptive evolution in mitochondrial genes, offering insights for phylogenetic and functional studies in Hydrangeaceae.

### 4.3. Mitogenome-Based Phylogenetic Analysis

Phylogenetic analysis in plants has advanced to utilize complete genomic data to reconstruct relationships among different species. Here, a phylogenetic tree was constructed based on the mt genomes of 35 species. Given the scarcity of publicly available mt genome data from other species within the same genus (*Platycrater*) or even the same family (Hydrangeaceae), we selected 33 closely related species from the Asteranae to construct the phylogenetic tree. *A. thaliana* (a member of the Rosanae) was chosen as the outgroup. The phylogenetic tree revealed that *H. macrophylla* and *P. arguta*, both within the family Hydrangeaceae, formed the closest clade, followed by *Eucommia ulmoides* (family Eucommiaceae) and *Rhododendron simsii* (family Ericaceae). The mt genome of *Hydrangea macrophyll* has a total length of 599,499 bp, which is shorter than that of *P. arguta*. The most widely cultivated Rhododendron species is *Rhododendron simsii* (Indoor azalea), its mt genome is 802,707 bp in length and contains 53 unique genes (33 protein-coding, 17 tRNA, and 3 rRNA genes) [45]. *Rhododendron delavayi,* another species in Rhododendron, shows a full-length mt genome spanning 1,009,263 bp, comprising 53 protein-coding genes, including 18 transfer RNA (tRNA) genes, 3 ribosomal RNA (rRNA) genes, and 32 protein-coding genes [46]. The mt genome of *R. delavayi* was the largest reported within the *Rhododendron* genus to date, and both of these two mt genomes are longer than the mt genome of *P. arguta*.

### 4.4. Inter-Organellar DNA Fragment Transfer Events

In this study, inter-organellar DNA fragment transfer was identified, with 31 cp-derived fragments migrating to the *P. arguta* mt genome, harboring 12 intact tRNA genes and partial sequences of 10 PCGs. The transferred tRNA genes exhibited preferential retention over PCGs, illustrating their conserved evolutionary role in mt genome plasticity. Characterization of the *P. arguta* mitochondrial genome enables comparative organellar phylogenomics, offering a critical counterpart to existing cp genome data for resolving evolutionary relationships within Hydrangeaceae [47]. This dual-organelle perspective may reveal mitochondrial–chloroplast sequence transfers and cytonuclear coevolution, key mechanisms driving plant adaptation [48]. Future studies should test whether the observed tRNA duplications reflect compensatory evolution between nuclear and organellar genomes.

## 5. Conclusions

In this study, we successfully sequenced and assembled the mt genome of *P. arguta* into two contigs, containing 37 PCGs, 27 tRNA genes, and three rRNA genes. The codon usage analysis revealed a pronounced A/U bias at the third codon position, with leucine and serine being the most frequent amino acids. A total of 687 RNA editing sites were predicted, predominantly altering amino acid hydrophobicity, and repeat sequences of different sizes were detected in the mt genome, including 160 SSRs, 26 tandem repeats, and 320 dispersed repeats. Ka/Ks analysis indicated purifying selection for most PCGs, while *atp4*, *ccmB*, *nad4*, and others showed signs of positive selection in specific species comparisons. Nucleotide diversity (Pi) values ranged from 0.01782 (*rps14*) to 0.10852 (*atp9*), highlighting conserved regions. Phylogenetic analysis placed *P. arguta* in Group V, closely related to *H. macrophylla*, with evolutionary ties to Eucommiaceae and Ericaceae. Additionally, 31 homologous fragments (41,871 bp) between mitochondrial and chloroplast genomes were identified, involving tRNA and rRNA gene transfers. This study provides critical insights into the *P. arguta* mt genome, offering a foundation for exploring genetic variation, evolutionary relationships, and molecular breeding in this species.

## Figures and Tables

**Figure 1 cimb-47-00521-f001:**
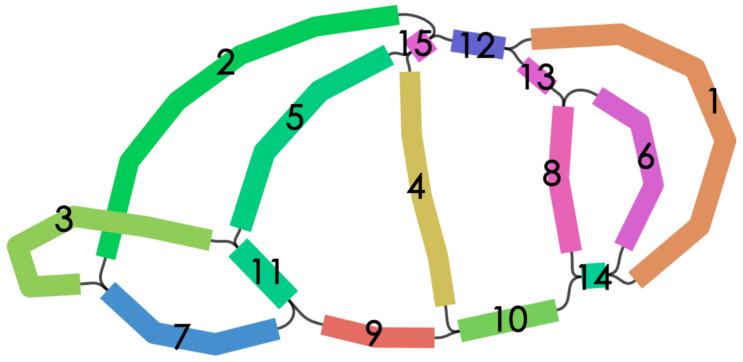
The graph model of the *P. arguta* mt genome. Chromosome 1 contains contigs 2, 3, 5, 7, 9, 11, and 15, and chromosome 2 contains contigs 1, 4, 6, 8, 10, 12, 13, 14, and 15.

**Figure 2 cimb-47-00521-f002:**
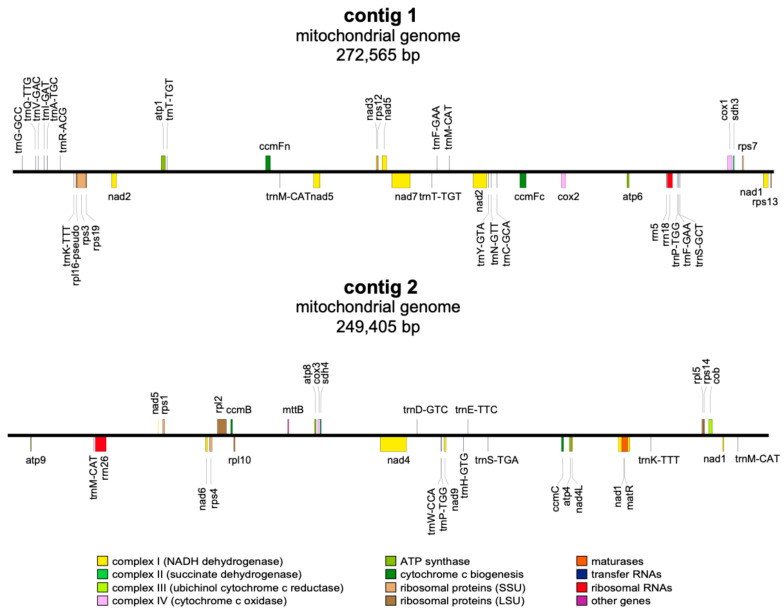
The maps of the *P. arguta* mt genomes. The forward-encoded genes in the linear display map are located above the line, and the reversely encoded genes are located below the line. Color-coding is used to distinguish genes of different functional classes.

**Figure 3 cimb-47-00521-f003:**
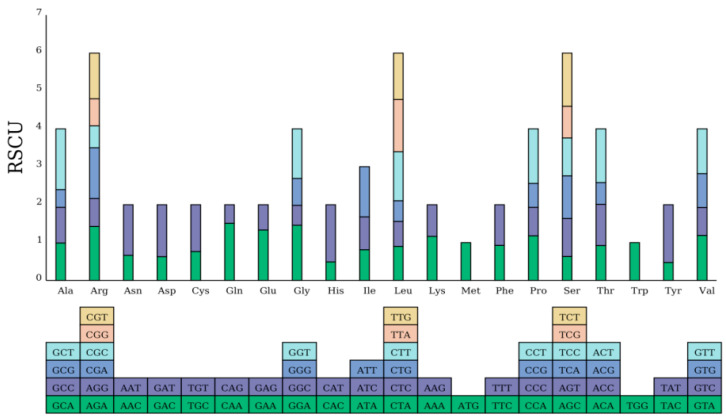
RSCU analysis of the *P. arguta* mt genome.

**Figure 4 cimb-47-00521-f004:**
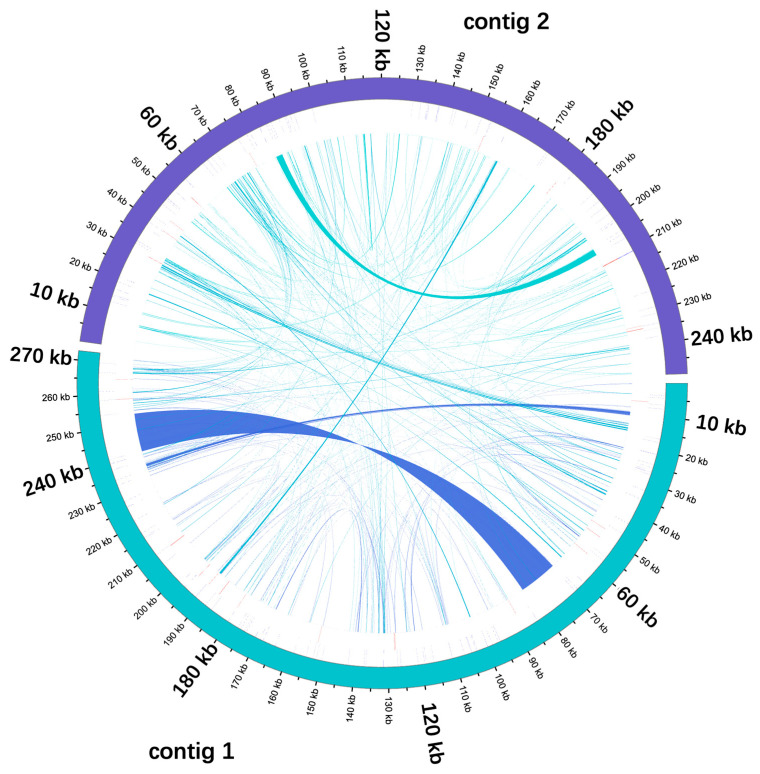
The distribution map of repetitive sequences on the mitogenome of *P. arguta*. The outermost circle represents the mt genome sequence, with inward concentric layers sequentially depicting SSRs, tandem repeats, and dispersed repeats.

**Figure 5 cimb-47-00521-f005:**
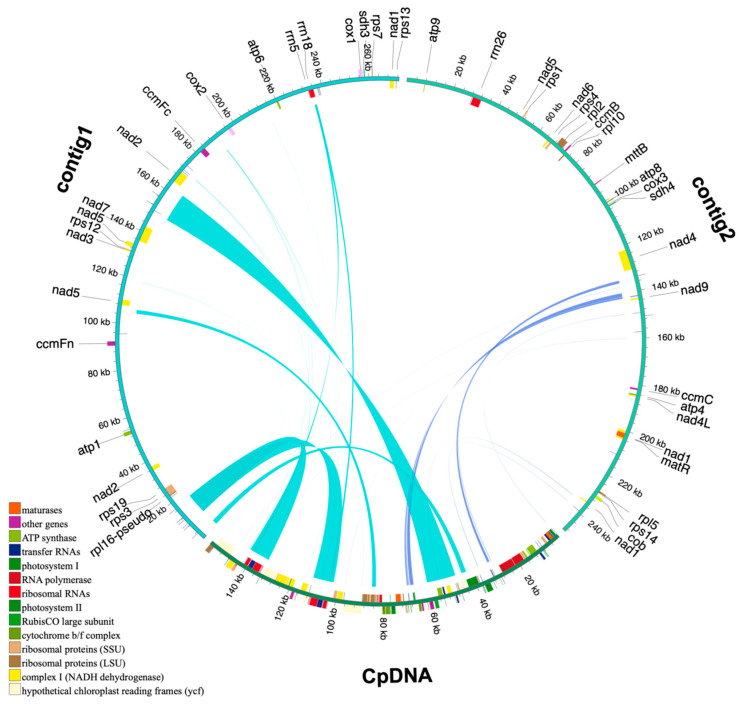
Transfer events in the cpgenome and mitogenome of *P. arguta*. Chloroplast sequences are labeled as CpDNA, while all other sequences correspond to mitochondrial DNA. Genes originating from the same complex are represented by identically colored blocks. Blocks in the outer ring and inner ring indicate genes located on the positive strand and negative strand, respectively. Homologous sequences are denoted by connecting lines at the central interface.

**Figure 6 cimb-47-00521-f006:**
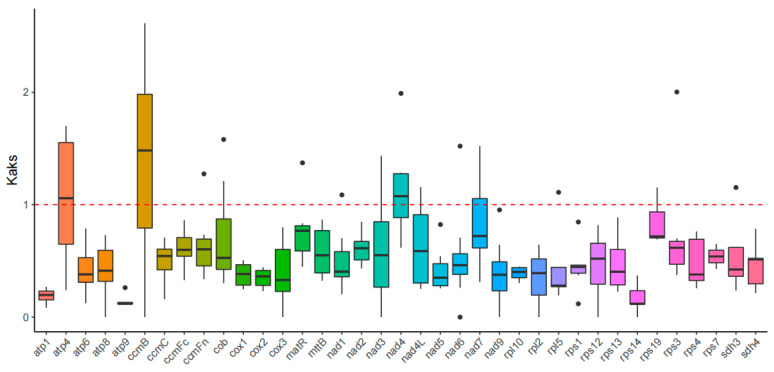
Ka/Ks ratios of 37 PCGs between *P. arguta* and eight species. The “X” axis shows the name of the PCGs, and the “Y” axis shows the Ka/Ks values. The rough line inside the rectangle represents the median, the upper and lower edges of the rectangle represent the upper and lower quartiles, and black dots represent outliers.

**Figure 7 cimb-47-00521-f007:**
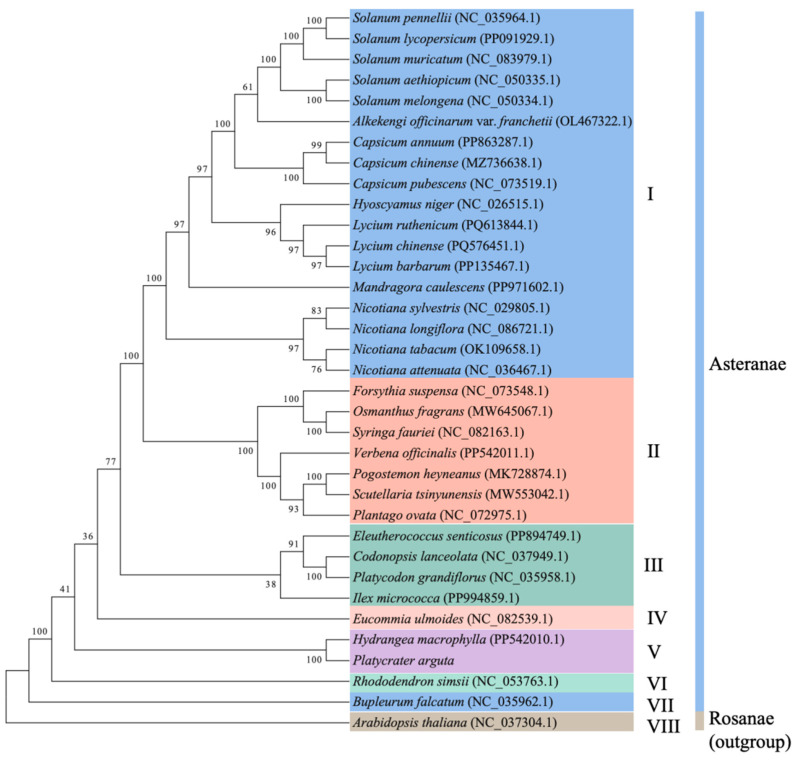
Maximum-likelihood phylogenetic tree based on 37 conserved PCGs among 35 species. *A*. *thaliana* was used as the outgroup. The 35 species were divided into eight groups (Group I–VIII). NCBI accession numbers are listed behind the scientific name.

**Table 1 cimb-47-00521-t001:** Gene composition in the mitogenome of the *P. arguta*.

Group of Genes		Gene Name
Core genes	ATP synthase	*atp1*, *atp4*, *atp6*, *atp8*, *atp9*
	Cytochrome c biogenesis	*ccmB*, *ccmC*, *ccmFc* *, *ccmFn*
	Ubichinol cytochrome c reductase	*Cob*
	Cytochrome c oxidase	*cox1*, *cox2* *, *cox3*
	Maturase	*matR*
	Transport membrance protein	*mttB*
	NADH dehydrogenase	*nad1* **, *nad2* **, *nad3*, *nad4* ***, *nad4L*, *nad5* ***, *nad6*, *nad7* ****, *nad9*
Variable genes	Large subunit of ribosome (LSU)	*rpl2*, *rpl10*, *rpl5*
	Small subunit of ribosome (SSU)	*rps1*, *rps3* *, *rps4*, *rps7*, *rps12*, *rps13*, *rps14*, *rps19*
	Succinate dehydrogenase	*sdh3*, *sdh4*
rRNA genes	Ribosomal RNAs	*rrn5*, *rrn18*, *rrn26*
tRNA genes	Transfer RNAs	*trnA-TGC*, *trnC-GCA*, *trnD-GTC*, *trnE-TTC*, *trnF-GAA* (×2), *trnG-GCC*, *trnH-GTG*, *trnI-GAT*, *trnK-TTT* (×2), *trnM-CAT* (×4), *trnN-GTT*, *trnP-TGG* (×2), *trnQ-TTG*, *trnR-ACG*, *trnS-GCT*, *trnS-TGA*, *trnT-TGT* (×2), *trnV-GAC*, *trnW-CCA*, *trnY-GTA*

* indicates one intron, ** indicate two introns, *** indicate three introns, and **** indicate four introns.

**Table 2 cimb-47-00521-t002:** Frequency of identified SSR motifs in *P. arguta* mitogenome.

Motif Type	Number of Repeats													Total	Proportion (%)
	3	4	5	6	7	8	9	10	11	12	13	14	15		
Monomer	-	-	-	-	-	-	-	26	12	8	2	1	0	49	30.63
Dimer	-	0	33	2	3	0	0	0	0	0	0	0	0	38	23.75
Trimer	-	9	2	0	0	0	0	0	0	0	0	0	0	11	6.87
Tetramer	51	1	0	0	0	0	0	0	0	0	0	0	0	52	32.5
Pentamer	7	0	0	0	0	0	0	0	0	0	0	0	1	8	5
Hexamer	2	0	0	0	0	0	0	0	0	0	0	0	0	2	1.25
Total	60	10	35	2	3	0	0	26	12	8	2	1	1	160	100

**Table 3 cimb-47-00521-t003:** Prediction of RNA editing sites in the *P. arguta* mt genome.

Type	RNA-Editing	Number	Percentage
hydrophilic–hydrophilic	CAC (H) => TAC (Y)	9	
	CAT (H) => TAT (Y)	22	
	CGC (R) => TGC (C)	12	10.77%
	CGT (R) => TGT (C)	31	
hydrophilic–hydrophobic	ACA (T) => ATA (I)	6	
	ACC (T) => ATC (I)	6	
	ACG (T) => ATG (M)	8	
	ACT (T) => ATT (I)	8	
	CGG (R) => TGG (W)	34	46.87%
	TCA (S) => TTA (L)	94	
	TCC (S) => TTC (F)	44	
	TCG (S) => TTG (L)	52	
	TCT (S) => TTT (F)	70	
hydrophilic–stop	CAA (Q) => TAA (X)	3	0.87%
	CGA (R) => TGA (X)	3	
hydrophobic–hydrophilic	CCA (P) => TCA (S)	13	
	CCC (P) => TCC (S)	13	
	CCG (P) => TCG (S)	4	8.15%
	CCT (P) => TCT (S)	26	
hydrophobic–hydrophobic	CCA (P) => CTA (L)	52	
	CCC (P) => CTC (L)	9	
	CCC (P) => TTC (F)	10	
	CCG (P) => CTG (L)	42	
	CCT (P) => CTT (L)	31	
	CCT (P) => TTT (F)	12	
	CTC (L) => TTC (F)	17	
	CTT (L) => TTT (F)	33	
	GCA (A) => GTA (V)	8	
	GCC (A) => GTC (V)	4	
	GCG (A) => GTG (V)	7	
	GCT (A) => GTT (V)	4	
	Total	687	100%

## Data Availability

Data are contained within the article or Appendix A.

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
