# Peer review of "Characterization and Analysis of the Complete Mitochondrial Genome of Platycrater arguta"

_cimb, 2025, doi:10.3390/cimb47070521_

Round 1
Reviewer 1 Report
Comments and Suggestions for Authors
Please see the attached file.

Reviewer 2 Report
Comments and Suggestions for Authors
The manuscript is interesting because it characterizes and analyzes, for the first time, the complete mitochondrial genome of Platycrater arguta, a species considered endangered and of conservation importance. The most relevant aspect of the study is precisely the use of various techniques to identify mitochondrial DNA, analyze it, and construct a phylogenetic tree to identify its evolutionary relationships.
The work is relevant to the Journal, because there is only one other similar study for the same species, but analyzes the complete chloroplast genome, and fortunately is reported in this manuscript. Therefore, it complements the information already reported by other authors for genomes from other important organelles.
Because they use multiple analyses, I consider their methodology well-structured and the results to be supported, so the discussion is consistent with what they have in their results. Perhaps they could highlight at the end of the discussion, what the use of knowing the mitochondrial genome will be and based on what is already known about the chloroplast genome, emphasize the possibility of making organellar phylogenetic relationships and sequence transfers between genomes. I think they have the information to do future study, not necessarily in this work, but they could highlight it.
Regarding the figures, I found them to be adequate and of good quality. Only in Figure 3 should the RSCU values be added for each bar and color. This way, the values obtained by the codons are clearly visible, improving the figure.
Other minor observations include:
The initial paragraph of materials should be corrected, as it lacks the hyphen to link the words that are interrupted by the line break.
Start with a capital letter (line 114).
Scientific name in italics on lines 255 and 332, 336.
Reviewer 3 Report
Comments and Suggestions for Authors
The manuscript entitled "Characterization and Analysis of the Complete Mitochondrial Genome of Platycrater arguta by Comparative Genomic Approaches" reports the complete sequencing, assembly, annotation, and comparative analysis of the mitochondrial (mt) genome of P. arguta, a rare and endangered Tertiary relict shrub of Hydrangeaceae. The authors utilize Illumina and Nanopore sequencing platforms to assemble the mitochondrial genome into two linear contigs, and subsequently conduct comprehensive genomic analyses including codon usage bias, repeat structure, RNA editing prediction, inter-organelle DNA transfer, nucleotide diversity, selective pressure (Ka/Ks), and phylogenetic placement. Given the scarcity of whole genomic data for P. arguta, the work advances understanding in an area that has received little attention and will serve as a valuable genomic resource for further studies in conservation genetics, evolution, and molecular breeding of Hydrangeaceae species.
The language throughout the manuscript is understandable but contains numerous grammatical issues and typographical errors。Consider removing “Comparative Genomic Approaches” unless the analysis includes genome-level synteny or broader organellar genome comparisons beyond P. arguta vs. H. macrophylla.
Line 273, the claim that nad4 alone has “687 RNA editing sites” appears to be a typographical error, as this exceeds the total number (687) stated across all PCGs.
Line 17, “The mt genome encode 37 protein-coding genes”, encodes
Round 2
Reviewer 1 Report
Comments and Suggestions for Authors
I think the authors should wait till they get the accession number from NCBI for the mitogenome submission. They must include the accession number in the manuscript before it can be accepted/published. NCBI submission ID is not acceptable is not acceptable in a manuscript. To improve the quality of the article the authors should incorporate my suggestion. Also, I doubt the 100% coverage stated in the authors' notes. Coverage is given in fold-values line 100X or 4000X. PacBio Genome coverage can be calculated using the formula: Number of subread Bases (mapped)/Genome Size. Additionally, Genome coverage (based on the program/software [e.g. HGAP] used for genome assembly, one can find the coverage value in the PacBio coverage report). I would strongly advice the authors to include the correct coverage value in the manuscript.
Other than this, I am OK with the edits made by the authors.
Reviewer 3 Report
Comments and Suggestions for Authors
The authors have addressed my comments and improved the manuscript that it is acceptable for publication.
Author Response
We sincerely appreciate the time and expertise you dedicated to identifying issues and providing recommendations, which have significantly strengthened the quality of this manuscript.